# Generalized linear mixed model approach for analyzing water, sanitation, and hygiene facilities in Bangladesh: Insights from BDHS 2022 data

**Mahmila Sanjana Mim****¤, Md. Lutfor Rahaman, Anamul Haque Sajib**\*

Department of Statistics, University of Dhaka, Dhaka, Bangladesh

¤ Current address: Southeast University, Dhaka, Bangladesh
\* sajibstat@du.ac.bd

## Abstract

This study explores the key determinants and barriers to effective WASH facilities in Bangladesh, which are crucial for mitigating health issues and ensuring equitable access. By analyzing the 2022 Bangladesh Demographic and Health Survey (BDHS) data and accounting for design clustering using a Generalized Linear Mixed Model (GLMM), this study's methodology demonstrates superior performance compared to conventional logistic regression, as supported by Akaike Information Criterion (AIC) and likelihood ratio test (LRT). The study found that basic handwashing facility was significantly linked to the household head's age, partner's education, media exposure, women's empowerment, wealth index, and maternal factors such as – age and education of mothers of under 5 children. Basic sanitation was associated with regional factors, the household head's sex and age, household size, partner's education, working status, wealth index, and maternal factors. Access to basic drinking water was largely driven by the wealth index, while combined WASH facilities were influenced by household head's sex and age, household size, partner's education, working status, media exposure, wealth index, and maternal characteristics. The findings indicate that addressing WASH challenges in Bangladesh requires an integrated, multi-dimensional policy approach that considers key socio-demographic and economic factors—a strategy essential for achieving 6th Sustainable Development Goal (SDG).

## Introduction

The United Nations adopted the Sustainable Development Goals (SDGs) in 2015 to urge everyone to work together to eradicate poverty, safeguard the environment, and achieve peace and prosperity for all by 2030. There are 17 SDGs, and they are interlinked. They acknowledge that one goal's impact depends on others' progress, and that development requires a harmonious approach among social, economic,

**Data availability statement:** Third party data was obtained for this study from DHS Program. Data may be requested from DHS Program after creating an account and submitting a concept note. More access information can be found on the DHS Program website (https://dhsprogram.com/data/Access-Instructions.cfm). The authors confirm that interested researchers would be able to access these data in the same manner as the authors. The authors also confirm that they had no special access privileges that others would not have.

**Funding:** The author(s) received no specific funding for this work.

**Competing interests:** The authors have declared that no competing interests exist.

and environmental dimensions. Among 17 SDGs, the 6th goal is related to clean water and sanitation, which aims to ensure the availability and sustainable management of water and sanitation for all [1]. Evidence demonstrates that water pollution and poor sanitation can cause many transmitted diseases and conditions such as bacillary diarrhoea, cholera, typhoid, polio, viral hepatitis A, acute respiratory infections etc. [2–8]. The World Health Organization (WHO) says that almost 2 million people die every year around the world as they do not have enough WASH facilities, and most of them are children [8]. Although impressive progress on improving access to safe water, sanitation, and hygiene (WASH) has been witnessed globally, the year 2020 saw 3.85 billion people without basic hygiene service, 1.7 billion people without basic water service, and 780 million people without improved toilets at their health care facility [9]. This indicates that many people's lives are affected by unmet needs of WASH facilities. Hence, lack of adequate access to WASH services can exert widespread influence on one's health, food availability, income and education.

Like many other developing countries, Bangladesh is also facing problems due to a lack of proper WASH facilities. According to the latest data from the Joint Monitoring Programme (JMP) of UNICEF and WHO, only 59% of the population in Bangladesh had safely managed drinking water in 2020, and only 39% had managed sanitation. Moreover, basic hygiene facilities had been found only among 58% of the population [10]. Poor WASH facilities in Bangladesh lead to various negative consequences. Therefore, it is necessary to improve the WASH facilities. The government should take proper steps to ensure the availability of basic WASH services. For this purpose, the factors which are hindering the availability of basic WASH facilities should be identified properly.

Considering the harmful effects of inadequate WASH facilities, this study aims to identify the factors contributing to insufficient WASH services in Bangladesh. For doing this, the Bangladesh Demographic and Health Survey (BDHS), 2022 data have been utilized. The BDHS delivers nationally representative data on numerous aspects of population health and demographics in Bangladesh. Since BDHS employed a two-stage stratified cluster sampling approach, data from respondents within the same cluster are likely correlated. To ensure consistent and efficient parameter estimates, this correlation must be accounted for during estimation. A generalized linear mixed model (GLMM) can be utilized to model such correlated data [11]. The GLMM approach is particularly well-suited for this study because it enables the simultaneous estimation of fixed effects, which capture the impact of individual-level determinants, and random effects, which account for variability at the cluster level. Additionally, employing GLMM yields robust and interpretable analyses that can directly support evidence-based policy recommendations to enhance WASH services in Bangladesh.

To the best of our knowledge, no existing literature has identified the key determinants and barriers to effective individual and combined basic WASH facilities in Bangladesh using the latest BDHS 2022 data. Motivated by this lack, our study has aimed to bridge the existing research gap.

## Data and methodology

### Data sources

In this study, the secondary data extracted from the BDHS 2022 have been used, which can be freely downloaded from the DHS website upon request. The BDHS 2022 was conducted under the authority of the National Institute of Population Research and Training (NIPORT), Ministry of Health and Family Welfare (MOHFW), Medical Education and Family Welfare Division. Mitra and Associates which is a private research agency, implemented this survey from June 2022 to December 2022. The survey employed a two-stage stratified sampling approach, selecting 675 enumeration areas (EAs)—237 urban and 438 rural—based on probability proportional to EA size, following specifications provided by ICF. After a complete household listing, a systematic sample of 45 households per EA was chosen to ensure statistically reliable demographic and health estimates across Bangladesh's eight divisions. Eligible women in these households answered core questions on background characteristics and reproductive history. Thirty households in each EA received a long individual questionnaire, while the remaining 15 households received a shorter version. Additionally, biomarker measurements, including height, weight, blood pressure, and blood glucose assessments, were conducted in a subset of households. One rural cluster in Cox's Bazar, Chittagong, was excluded due to security concerns. As a result, the design resulted in the selection of 30,330 households, with 19,709 from rural and 10,665 from urban areas [12]. For this study, missing values were assessed, and cases with missing data in key variables were excluded to uphold methodological rigor and maintain the integrity of the statistical results, resulting in a final sample of 15,610 observations. The sample selection procedure is shown in Fig 1.

### Study variables

**Dependent variables.** The study has focused on household access to individual and combined basic WASH services. According to the WHO/UNICEF Joint Monitoring Programme (JMP) guidelines, these services were categorized into different levels: basic, limited, unimproved, and no service (Table 1 and Table 2) [13].

In order to create dependent variables (individual basic WASH facilities — basic handwashing facility, basic sanitation facility and basic drinking water facility) by following the guidelines of WHO/UNICEF Joint Monitoring Programme (JMP), a dichotomization has been performed:

$$Individual\ Basic\ WASH\ Facilities = \begin{cases} Yes = 1, & when\ the\ service\ level\ is\ basic \\ No = 0, & otherwise \end{cases}$$

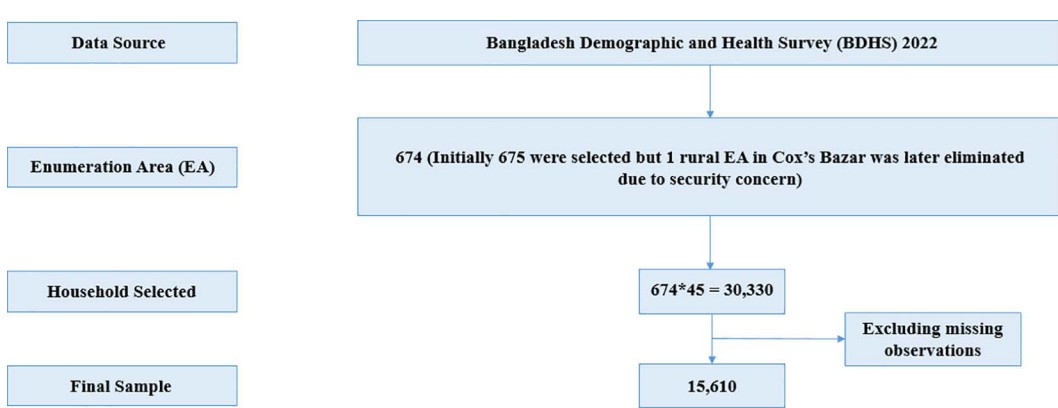

**Fig 1. Flowchart of sample selection from the BDHS 2022 dataset.**

**Table 1. WHO/UNICEF JMP ladder for WASH services.**

| Service level | Water | Sanitation | Hygiene |
|---|---|---|---|
| Basic | Drinking water from an improved source, provided collection time is not more than 30 min for a round trip, including queuing | Use of improved facilities that are not shared with other households | Availability of a handwashing facility on premises with soap and water |
| Limited | Drinking water from an improved source for which collection time exceeds 30 min for a round trip, including queuing | Use of improved facilities shared between two or more households | Availability of a handwashing facility on premises without soap and water |
| Unim-proved | Drinking water from an unprotected dug well or unprotected spring | Use of pit latrines without a slab or plat-form, hanging latrines or bucket latrines | Not applicable |
| No service | Surface water | Open defecation | No handwashing facility on premise |

**Table 2. JMP classification of improved/unimproved water and sanitation facility type.**

| Facility types | Water | Sanitation |
|---|---|---|
| Improved facilities | Piped supplies• Tap water in the dwelling, yard or plot<br>• Public standposts | Network sanitation• Flush and pour flush toilets connected to sewers |
| | Non-piped supplies | On-site sanitation |
| | • Boreholes/tubewells<br>• Protected wells and springs<br>• Rainwater<br>• Packaged water, including bottled water and sachet water<br>• Delivered water, including tanker trucks and small carts | • Flush and pour flush toilets or latrines connected to septic tanks or pit<br>• Ventilated improved pit latrines<br>• Pit latrines with slabs<br>• Composting toilets, including twin pit latrines and container-based system |
| Unimproved facilities | Non-piped supplies | On-site sanitation |
| | • Unprotected wells and springs | • Pit latrines without slabs<br>• Hanging latrines<br>• Bucket latrines |

While constructing the basic drinking water variable, the collection time for drinking water from an improved source has not been considered because of extensive missing data. Finally, the last binary dependent variable has been generated for households that combined all three individual basic WASH facilities, termed as combined basic WASH facilities. A household has been considered to have access to the combined basic WASH facilities only if all three basic facilities are available. That is,

$$\text{Combined Basic WASH Facilities} = \begin{cases} Yes = 1, & \text{all of three basic services are available} \\ No = 0, & \text{otherwise} \end{cases}$$

**Covariates.** Based on literature reviews [14–28], the covariates have been considered in this study: division (Barisal, Chittagong, Dhaka, Khulna, Mymensingh, Rajshahi, Rangpur and Sylhet), place of residence (urban and rural), sex of household head (male and female), age of household head ($< 30, 30-39, 40-49, 50-59$ and $\geq 60$), household size ($\leq 5$ and $> 5$), partner's education level (no education, primary, secondary and higher), working status (yes and no), media exposure (exposed and non-exposed), migration (migrant and non-migrant), women empowerment (yes and no), wealth index (poor, middle and rich), children aged 5 and under in household (yes and no), mother's age of under 5 children ($\leq 30$ and $> 30$), mother's education of under 5 children (no education, primary, secondary and higher). Here, the variable media exposure has been defined by categorizing the respondents who read magazines or newspapers, listen to the radio, or watch television once a week. The migration status has been determined based on whether individuals have

resided in their current homes for less than two years. The variable women empowerment has been generated based on active participation of women in four types of decisions: about their own health care, major household purchases, visiting family or relatives, and decisive actions on money earned by their husbands. If a woman is involved in any of these four decisions, she has been placed into category: yes; otherwise, she has been placed into category: no. Though the dataset originally divides the variable wealth index into five categories: poorest, poorer, middle, richer and richest, for this study, these have been consolidated into three groups: the poor category has been formed by combining the poorest and poorer groups, the middle category has remained unchanged and the rich category has been created by merging the richer and richest groups.

## Statistical analyses

Prior to analysis, outliers were identified using statistical techniques and visualized through boxplots, with values exceeding three standard deviations from the mean carefully evaluated. Categorical variables were appropriately coded to facilitate accurate analysis. The study variables have been summarized using univariate analysis, such as frequency distribution. The Chi-Square test has been used as a bivariate analysis to measure the connection between the response variable and the categorical explanatory variables. Finding the explanatory variables that are significantly associated with the response variable is one of the primary goals of bivariate analysis. Significant explanatory variables identified in bivariate analysis have been regarded as such in multivariate analysis. All preprocessing and analytical procedures were carried out using Statistical Package for Social Science (SPSS) software version 20 and R Programming.

**Generalized Linear Mixed Model (GLMM).** To take into account the correlation between the observations inside each cluster, the Generalized Linear Mixed Model (GLMM) has been used in this investigation. In contrast to the Generalized Linear Model (GLM), random effects have been incorporated alongside fixed effects to get estimates of the regression parameters that are more accurate. GLMMs extend GLMs to accommodate correlated responses within the same group, which contravenes the independence assumption. GLMM is regarded as a superior method for analyzing data that demonstrates a clustering effect [29].

Let, $y_{ip}$ be the $p^{th}$ individual from the $i^{th}$ cluster where $i = 1, 2, \ldots, q$ and, $p = 1, 2, \ldots, n_i$. Let, $x_{ip}$ be a vector of covariates for $p^{th}$ individual from $i^{th}$ cluster related with fixed effect parameter denoted by $\beta$ as variation may exist in a number of subjects per cluster. Also, let, $u$ be a $(q \times 1)$ vector of random effects associated with $q$ clusters and $z_{ip}$ be a unique vector of dimensions $(q \times 1)$, which is composed entirely of zeros except for a single entry of 1 at the $i^{th}$ position where, $i = 1, 2, \ldots, q$. One may write,

$$\mu_{ip} = E\left(y_{ip}|u_i\right),$$

where $u_i$ is the random effect of the $i^{th}$ cluster. The linear predictor in GLMM has the following form [11],

$$g\left(\mu_{ip}\right) = x_{ip}{}'\beta + z_{ip}{}'u.$$

For a continuous response variable,

$$y_{ip} = x_{ip}{}'\beta + z_{ip}{}'u + e_{ip},$$

which can be written in matrix notation as,

$$Y = X\beta + Zu + e,$$

here $Y$ denotes $n \times 1$ column vector, $X$ denotes $n \times k$ matrix of covariates, $\beta$ is $k \times 1$ vector of coefficients of fixed effects, $Z$ is $n \times q$ design matrix of $q$ random effects, $u$ denotes $q \times 1$ vector of random effects associated with q clusters and the $n \times 1$ vector of residuals is denoted by $e$ where, $n = \sum_{i=1}^{q} n_i$. $u$ follows a normal distribution with mean 0 and covariance matrix, $\Sigma$ [16].

Because of introducing a random slope in the model, the number of columns in the design matrix, $Z$, will be doubled, although the number of rows will remain the same. Hence, the covariance matrix, $\Sigma$, will become a $(2 \times 2)$ matrix as follows,

$$\Sigma = \begin{pmatrix} \sigma^2_{intercept} & \sigma^2_{intercept,slope} \\ \sigma^2_{slope,intercept} & \sigma^2_{slope} \end{pmatrix}$$

In this study, only the random intercept term is being considered, which turns $\Sigma$ into a scalar. In this case, the variance-covariance matrix of residuals is,

$$Var(e_{ip}) = \sigma^2_e.$$

In this structure, it is assumed that the variance of the residual is homogeneous for all observations and also that they are independent of each other. The between-group (or cluster) variance is denoted by $\sigma^2_u$, and the within-group variance is denoted by $\sigma^2_e$. The likelihood function for the individuals associated with $i^{th}$ cluster is,

$$L_i\left((\beta, \sigma^2_u)\,|x_{ip}, u_i\right) = f(y_i|x_{ip}, u_i) = \prod_{p=1}^{n_i} f(y_i|x_{ip}, u_i)$$

The marginal likelihood function is [30],

$$L\left((\beta, \sigma^2_u)\,|x_{ip}\right) = \int_{-\infty}^{\infty} \left[\prod_{i=1}^{q} L_i\left((\beta, \sigma^2_u)\,|u_i\right)\right] g(u_i)\, du_i.$$

It is quite impossible to have an explicit solution from this expression. For this reason, some techniques have been utilized for approximation to obtain the maximum likelihood estimates. Some of these approximation techniques are: Gauss-Hermite quadrature, Laplace approximation, Penalized quasi-likelihood method [31–33].

**GLMM for binary response: Mixed-effect logistic regression model.** When the response variable is binary, employing a mixed-effect logistic regression within the Generalized Linear Mixed Models (GLMM) framework is suitable for analyzing such data. The link function under this model is of the following form [34],

$$g(\mu_{ip}) = log\left(\frac{\mu_{ip}}{1 - \mu_{ip}}\right),$$

and,

$$log\left(\frac{\mu_{ip}}{1 - \mu_{ip}}\right) = x_{ip}{}'\beta + z_{ip}{}'u,$$

where,

$$u_i \sim N(0, \Sigma)$$

Because of considering only the random intercept term in this study, $\Sigma$ can be replaced by $\sigma_u^2$,

$$u_i \sim N\left(0, \sigma_u^2\right)$$

The conditional probability of $p^{th}$ observation from $i^{th}$ cluster given the value of the covariate of that observation and the random cluster effect is given by,

$$\mu_{ip} = E\left[Y_{ip}|x_{ip}, u_i\right] = Pr\left[Y_{ip} = 1|x_{ip}, u_i\right] = \frac{e^{\beta' x_{ip} + u_i}}{1 + e^{\beta' x_{ip} + u_i}}$$

**Intra-cluster correlation (ICC).** Intra-cluster correlation is a statistical measure that is used to measure how similar the outcomes are that belong to the same cluster [35]. For a binary response variable, the model can be written as,

$$Y_{ip} = \mu + u_i + e_{ip}; \qquad u_i = z_{ip}'u,$$

where, $\mu$ is fixed, $u_i$ follows a normal distribution with zero mean and variance $\sigma^2$, $e_{ip}$ has a standard logistic distribution with zero mean and variance $\sigma_e^2$. If the response variable is binary type, the within-cluster variance $\sigma_e^2$ is replaced by $\frac{\pi^2}{3}$ for convenience [36]. Then the intra-cluster correlation (ICC) for a binary outcome denoted by $\rho$ can be defined as,

$$\rho = \frac{\sigma_u^2}{\sigma_u^2 + \frac{\pi^2}{3}}$$

Since, the response variables of our study (individual and combined basic WASH facilities) are binary, mixed-effect logistic regression model within the GLMM framework has been employed for analyzing data. Based on the likelihood ratio test (LRT) and AIC values, we have compared the performance of the fixed-effect logistic regression model and the mixed-effect logistic regression model. AIC has the following mathematical form:

$$AIC = -2\ln(L) + 2\gamma,$$

where $n$ is the number of observations and $\gamma$ is the number of parameters to be estimated. The model with the lowest AIC is typically preferred since it offers a better fit. All the analyses of our study have been performed by using the Statistical Package for Social Science (SPSS) software version 20 and R version 4.1.1.

## Results

### Univariate analysis

Table 3 presents the frequency, valid percentage, and cumulative percentage of each study variable. It shows that 59.7% of households had access to basic handwashing facility, 62.7% to basic sanitation, 98.7% to basic drinking water, and 45.0% to combined WASH facilities. The highest number of respondents was from Dhaka (15.7%), and the lowest from Sylhet (10.2%). A larger proportion resided in rural areas (64.6%) compared to urban areas (35.4%). Most household heads were male (88.6%), with 28.9% aged 40–49 years and 13.5% aged under 30. The majority had a household size of 5 or fewer (74.7%). Among respondents' partners, 23.0% were illiterate, 31.0% had secondary education (highest), and 16.6% had higher education (lowest). Additionally, 33.2% of respondents were currently working, 58.4% had media exposure, and 90.6% were non-migrants. Women empowerment stood at 89.1%. Of the 15,610 respondents, 37.9% were from poor families, 19.8% from middle-class families, and 42.3% from rich families. Nearly half (47.9%) had children aged 5 or

**Table 3. Univariate analysis of dependent and independent variables.**

| Variable | Frequency, n | Valid Percent, % | Cumulative Percent, % |
|---|---|---|---|
| **Basic Handwashing Facility** | | | |
| Yes | 9324 | 59.7 | 59.7 |
| No | 6286 | 40.3 | 100.0 |
| **Basic Sanitation Facility** | | | |
| Yes | 9784 | 62.7 | 62.7 |
| No | 5826 | 37.3 | 100.0 |
| **Basic Drinking Water Facility** | | | |
| Yes | 15410 | 98.7 | 98.7 |
| No | 200 | 1.3 | 100.0 |
| **Combined WASH Facilities** | | | |
| Yes | 7018 | 45.0 | 45.0 |
| No | 8592 | 55.0 | 100.0 |
| **Division** | | | |
| Barishal | 1662 | 10.6 | 10.6 |
| Chattogram | 2239 | 14.3 | 25.0 |
| Dhaka | 2450 | 15.7 | 40.7 |
| Khulna | 2060 | 13.2 | 53.9 |
| Mymensingh | 1690 | 10.8 | 64.7 |
| Rajshahi | 2056 | 13.2 | 77.9 |
| Rangpur | 1860 | 11.9 | 89.8 |
| Sylhet | 1593 | 10.2 | 100.0 |
| **Place of Residence** | | | |
| Urban | 5528 | 35.4 | 35.4 |
| Rural | 10082 | 64.6 | 100.0 |
| **Sex of Household Head** | | | |
| Male | 13827 | 88.6 | 88.6 |
| Female | 1783 | 11.4 | 100.0 |
| **Age of Household Head** | | | |
| <30 | 2115 | 13.5 | 13.5 |
| 30-39 | 3710 | 23.8 | 37.3 |
| 40-49 | 4506 | 28.9 | 66.2 |
| 50-59 | 3002 | 19.2 | 85.4 |
| ≥60 | 2277 | 14.6 | 100.0 |
| **Household Size** | | | |
| ≤5 | 11653 | 74.7 | 74.7 |
| >5 | 3957 | 25.3 | 100.0 |
| **Partner's Education Level** | | | |
| No Education | 3594 | 23.0 | 23.0 |
| Primary | 4583 | 29.4 | 52.4 |
| Secondary | 4836 | 31.0 | 83.4 |
| Higher | 2597 | 16.6 | 100.0 |
| **Working Status** | | | |
| Yes | 5177 | 33.2 | 33.2 |
| No | 10433 | 66.8 | 100.0 |

*(Continued)*

**Table 3.** (Continued)

| Variable | Frequency, n | Valid Percent, % | Cumulative Percent, % |
|---|---|---|---|
| **Media Exposure** | | | |
| Exposed | 9112 | 58.4 | 58.4 |
| Non-exposed | 6498 | 41.6 | 100.0 |
| **Migration** | | | |
| Migrant | 1464 | 9.4 | 9.4 |
| Non-migrant | 14146 | 90.6 | 100.0 |
| **Woman Empowerment** | | | |
| Yes | 13912 | 89.1 | 89.1 |
| No | 1698 | 10.9 | 100.0 |
| **Wealth Index** | | | |
| Poor | 5912 | 37.9 | 37.9 |
| Middle | 3089 | 19.8 | 57.7 |
| Rich | 6609 | 42.3 | 100.0 |
| **Children Aged 5 and Under in Household** | | | |
| Yes | 7483 | 47.9 | 47.9 |
| No | 8127 | 52.1 | 100.0 |
| **Mother's Age of Under 5 Children** | | | |
| ≤30 | 6250 | 40.0 | 40.0 |
| >30 | 9360 | 60.0 | 100.0 |
| **Mother's Education of Under 5 Children** | | | |
| No Education | 2233 | 14.3 | 14.3 |
| Primary | 4430 | 28.4 | 42.7 |
| Secondary | 5327 | 34.1 | 76.8 |
| Higher | 3620 | 23.2 | 100.0 |

under. Among these mothers, 40.0% were aged 30 or younger, and 60.0% were older than 30. Educational levels among these mothers were: 14.3% illiterate, 28.4% primary, 34.1% secondary, and 23.2% higher education.

## Bivariate analysis

Bivariate analysis represents the most fundamental form of quantitative analysis used to uncover the empirical correlation between two distinct variables. Essentially, it seeks to ascertain the presence of a link between the two variables. In the realm of bivariate analysis, the technique of cross-tabulation is employed to examine the connection between two variables through their arrangement in a tabular format. Results obtained from bivariate analysis for individual and combined basic WASH facilities are presented in Table 4.

Table 4 reveals that access to basic handwashing facility was significantly associated with the place of residence, household head's sex and age, household size, partner's education level, working status, media exposure, migration status, women empowerment, wealth index, presence of children aged 5 and under, and the mother's age and education of these children. Division, place of residence, sex and age of the household head, household size, partner's education level, working status, media exposure, wealth index, presence of children aged 5 and under, mother's age, and education of these children were all significantly correlated with access to basic sanitation facilities, as shown in Table 4. Moreover, Table 4 shows a significant relationship between access to basic drinking water facility and the partner's education level, employment position,

**Table 4. Bivariate frequency distribution of individual and combined basic WASH facilities among the different categories of selected covariates along with p-value.**

| Explanatory Variables | Handwashing Facility | | | Sanitation Facility | | | Drinking Water Facility | | | Combined WASH Facilities | | |
|---|---|---|---|---|---|---|---|---|---|---|---|---|
| | Yes, n(%) | No, n(%) | p-value | Yes, n(%) | No, n(%) | p-value | Yes, n(%) | No, n(%) | p-value | Yes, n(%) | No, n(%) | p-value |
| **Division** | | | 0.153 | | | < 0.001 | | | 0.496 | | | 0.151 |
| Barisal | 861 (51.8) | 801 (48.2) | | 1182 (71.1) | 480 (28.9) | | 1643 (98.9) | 19 (1.1) | | 698 (42.0) | 964 (58.0) | |
| Chittagong | 1417 (63.3) | 822 (36.7) | | 1472 (65.7) | 767 (34.3) | | 2209 (98.7) | 30 (1.3) | | 1112 (49.7) | 1127 (50.3) | |
| Dhaka | 1536 (62.7) | 914 (37.3) | | 1438 (58.7) | 1012 (41.3) | | 2429 (99.1) | 21 (0.9) | | 1116 (45.6) | 1334 (54.4) | |
| Khulna | 1275 (61.9) | 785 (38.1) | | 1348 (65.4) | 712 (34.6) | | 1991 (96.7) | 69 (3.3) | | 978 (47.5) | 1082 (52.5) | |
| Mymensingh | 862 (51.0) | 828 (49.0) | | 937 (55.4) | 753 (44.6) | | 1690 (100.0) | 0 (0.0) | | 605 (35.8) | 1085 (64.2) | |
| Rajshahi | 1258 (61.2) | 798 (38.8) | | 1333 (64.8) | 723 (35.2) | | 2051 (99.8) | 5 (0.2) | | 969 (47.1) | 1087 (52.9) | |
| Rangpur | 1203 (64.7) | 657 (35.3) | | 1085 (58.3) | 775 (41.7) | | 1860 (100.0) | 0 (0.0) | | 841 (45.2) | 1019 (54.8) | |
| Sylhet | 912 (57.3) | 681 (42.7) | | 989 (62.1) | 604 (37.9) | | 1537 (96.5) | 56 (3.5) | | 699 (43.9) | 894 (56.1) | |
| **Place of Residence** | | | < 0.001 | | | < 0.001 | | | 0.473 | | | < 0.001 |
| Urban | 3934 (71.2) | 1594 (28.8) | | 3725 (67.4) | 1803 (32.6) | | 5462 (98.8) | 66 (1.2) | | 3155 (57.1) | 2373 (42.9) | |
| Rural | 5390 (53.5) | 4692 (46.5) | | 6059 (60.1) | 4023 (39.9) | | 9948 (98.7) | 134 (1.3) | | 3863 (38.3) | 6219 (61.7) | |
| **Sex of Household Head** | | | 0.005 | | | < 0.001 | | | 0.630 | | | < 0.001 |
| Male | 8204 (59.3) | 5623 (40.7) | | 8585 (62.1) | 5242 (37.9) | | 13652 (98.7) | 175 (1.3) | | 6139 (44.4) | 7688 (55.6) | |
| Female | 1120 (62.8) | 663 (37.2) | | 1199 (67.2) | 584 (32.8) | | 1758 (98.6) | 25 (1.4) | | 879 (49.3) | 904 (50.7) | |
| **Age of Household Head** | | | < 0.001 | | | < 0.001 | | | 0.735 | | | < 0.001 |
| <30 | 1085 (51.3) | 1030 (48.7) | | 888 (42.0) | 1227 (58.0) | | 2076 (98.2) | 39 (1.8) | | 619 (70.7) | 1496 (29.3) | |
| 30-39 | 2179 (58.7) | 1531 (41.3) | | 2126 (57.3) | 1584 (42.7) | | 3678 (99.1) | 32 (0.9) | | 1581 (57.4) | 2129 (42.6) | |
| 40-49 | 2744 (60.9) | 1762 (39.1) | | 2968 (65.9) | 1538 (34.1) | | 4451 (98.8) | 55 (1.2) | | 2119 (53.0) | 2387 (47.0) | |
| 50-59 | 1905 (63.5) | 1097 (36.5) | | 2141 (71.3) | 861 (28.7) | | 2965 (98.8) | 37 (1.2) | | 1537 (48.8) | 1465 (51.2) | |
| ≥60 | 1411 (62.0) | 866 (38.0) | | 1661 (72.9) | 616 (27.1) | | 2240 (98.4) | 37 (1.6) | | 1162 (51.0) | 1115 (49.0) | |
| **Household Size** | | | 0.021 | | | < 0.001 | | | 0.482 | | | < 0.001 |
| ≤5 | 6899 (59.2) | 4754 (40.8%) | | 7003 (60.1) | 4650 (39.9) | | 11508 (98.8) | 145 (1.2) | | 5059 (43.4) | 6594 (56.6) | |
| >5 | 2425 (61.3) | 1532 (38.7%) | | 2781 (70.3) | 1176 (29.7) | | 3902 (98.6) | 55 (1.4) | | 1959 (49.5) | 1998 (50.5) | |

*(Continued)*

| Explanatory Variables | Handwashing Facility | | | Sanitation Facility | | | Drinking Water Facility | | | Combined WASH Facilities | | |
|---|---|---|---|---|---|---|---|---|---|---|---|---|
| | Yes, n(%) | No, n(%) | p-value | Yes, n(%) | No, n(%) | p-value | Yes, n(%) | No, n(%) | p-value | Yes, n(%) | No, n(%) | p-value |
| **Partner's Education Level** | | | < 0.001 | | | < 0.001 | | | < 0.001 | | | < 0.001 |
| No Education | 1696 (47.2) | 1898 (52.8) | | 1839 (51.2) | 1755 (48.8) | | 3532 (98.3) | 62 (1.7) | | 1068 (29.7) | 2526 (70.3) | |
| Primary | 2331 (50.9) | 2252 (49.1) | | 2497 (54.5) | 2086 (45.5) | | 4516 (98.5) | 67 (1.5) | | 1577 (34.4) | 3006 (65.6) | |
| Secondary | 3100 (64.1) | 1736 (35.9) | | 3216 (66.5) | 1620 (33.5) | | 4777 (98.8) | 59 (1.2) | | 2383 (49.3) | 2453 (50.7) | |
| Higher | 2197 (84.6) | 400 (15.4) | | 2232 (85.9) | 365 (14.1) | | 2585 (99.5) | 12 (0.5) | | 1990 (76.6) | 607 (23.4) | |
| **Working Status** | | | < 0.001 | | | < 0.001 | | | 0.039 | | | < 0.001 |
| Yes | 2916 (56.3) | 2261 (43.7) | | 3040 (58.7) | 2137 (41.3) | | 5097 (98.5) | 80 (1.5) | | 2077 (40.1) | 3100 (59.9) | |
| No | 6408 (61.4) | 4025 (38.6) | | 6744 (64.6) | 3689 (35.4) | | 10313 (98.8) | 120 (1.2) | | 4941 (47.4) | 5492 (52.6) | |
| **Media Exposure** | | | < 0.001 | | | < 0.001 | | | < 0.001 | | | < 0.001 |
| Exposed | 6096 (66.9) | 3016 (33.1) | | 6165 (67.7) | 2947 (32.3) | | 9040 (99.2) | 72 (0.8) | | 4826 (53.0) | 4286 (47.0) | |
| Non-exposed | 3228 (49.7) | 3270 (50.3) | | 3619 (55.7) | 2879 (44.3) | | 6370 (98.0) | 128 (2.0) | | 2192 (33.7) | 4306 (66.3) | |
| **Migration** | | | 0.001 | | | 0.376 | | | 0.359 | | | 0.001 |
| Migrant | 932 (63.7) | 532 (36.3) | | 902 (61.6) | 562 (38.4) | | 1449 (99.0) | 15 (1.0) | | 718 (49.0) | 746 (51.0) | |
| Non-migrant | 8392 (59.3) | 5754 (40.7) | | 8882 (62.8) | 5264 (37.2) | | 13961 (98.7) | 185 (1.3) | | 6300 (44.5) | 7846 (55.5) | |
| **Woman Empowerment** | | | < 0.001 | | | 0.779 | | | 0.019 | | | 0.002 |
| Yes | 8398 (60.4) | 5514 (39.6) | | 8725 (62.7) | 5187 (37.3) | | 13744 (98.8) | 168 (1.2) | | 6316 (45.4) | 7596 (54.6) | |
| No | 926 (54.5) | 772 (45.5) | | 1059 (62.4) | 639 (37.6) | | 1666 (98.1) | 32 (1.9) | | 702 (41.3) | 996 (58.7) | |
| **Wealth Index** | | | < 0.001 | | | < 0.001 | | | < 0.001 | | | < 0.001 |
| Poor | 2171 (36.7) | 3741 (63.3) | | 2523 (42.7) | 3389 (57.3) | | 5795 (98.0) | 117 (2.0) | | 1101 (18.6) | 4811 (81.4) | |
| Middle | 1722 (55.7) | 1367 (44.3) | | 1846 (59.8) | 1243 (40.2) | | 3061 (99.1) | 28 (0.9) | | 1191 (38.6) | 1898 (61.4) | |
| Rich | 5431 (82.2) | 1178 (17.8) | | 5415 (81.9) | 1194 (18.1) | | 6554 (99.2) | 55 (0.8) | | 4726 (71.5) | 1883 (28.5) | |
| **Children Aged 5 and Under in Household** | | | < 0.001 | | | < 0.001 | | | 0.682 | | | < 0.001 |
| Yes | 4399 (58.8) | 3084 (41.2) | | 4465 (59.7) | 3018 (40.3) | | 7390 (98.8) | 93 (1.2) | | 3212 (42.9) | 4271 (57.1) | |
| No | 4925 (60.6) | 3202 (39.4) | | 5319 (65.4) | 2808 (34.6) | | 8020 (98.7) | 107 (1.3) | | 3806 (46.8) | 4321 (53.2) | |

*(Continued)*

**Table 4.** (Continued)

| Explanatory Variables | Handwashing Facility | | | Sanitation Facility | | | Drinking Water Facility | | | Combined WASH Facilities | | |
|---|---|---|---|---|---|---|---|---|---|---|---|---|
| | Yes, n(%) | No, n(%) | p-value | Yes, n(%) | No, n(%) | p-value | Yes, n(%) | No, n(%) | p-value | Yes, n(%) | No, n(%) | p-value |
| **Mother's Age of Under 5 Children** | | | < 0.001 | | | < 0.001 | | | 0.893 | | | < 0.001 |
| ≤30 | 3551 (56.8) | 2699 (43.2) | | 3411 (54.6) | 2839 (45.4) | | 6169 (98.7) | 81 (1.3) | | 2433 (38.9) | 3817 (61.1) | |
| >30 | 5773 (61.7) | 3587 (38.3) | | 6373 (68.1) | 2987 (31.9) | | 9241 (98.7) | 119 (1.3) | | 4585 (49.0) | 4775 (51.0) | |
| **Mother's Education of Under 5 Children** | | | < 0.001 | | | < 0.001 | | | < 0.001 | | | < 0.001 |
| No Education | 1045 (46.8) | 1188 (53.2) | | 1153 (51.6) | 1080 (48.4) | | 2189 (97.6) | 53 (2.4) | | 682 (30.5) | 1551 (69.5) | |
| Primary | 2172 (49.0) | 2258 (51.0) | | 2392 (54.0) | 2038 (46.0) | | 4369 (98.6) | 61 (1.4) | | 1439 (32.5) | 2991 (67.5) | |
| Secondary | 3192 (59.9) | 2135 (40.1) | | 3331 (62.5) | 1996 (37.5) | | 5267 (98.9) | 60 (1.1) | | 2374 (44.6) | 2953 (55.4) | |
| Higher | 2915 (80.5) | 705 (19.5) | | 2908 (80.3) | 712 (19.7) | | 3594 (99.3) | 26 (0.7) | | 2523 (69.7) | 1097 (30.3) | |

media exposure, women's empowerment, wealth index, and the mother's education of children under 5. It is also found from Table 4 that access to combined WASH facilities were significantly associated with the place of residence, sex and age of the household head, household size, partner's education level, working status, media exposure, migration status, women empowerment, wealth index, presence of children aged 5 and under, and the mother's age and education of these children. All of these associations have been found significant as the corresponding *p*-values are less than $\alpha = 0.05$.

## Generalized linear mixed model: Analyzing WASH facilities

To capture the clustering effect adequately, a mixed-effect logistic regression model under the GLMM framework has been utilized. In this model, a random effect component has been incorporated for each cluster. This assumes that the baseline odds of the event occurring are consistent for all respondents within the same cluster, while the odds vary from one cluster to another. Four separate sets of GLMMs for binary responses have been considered for basic handwashing facility, sanitation facility, drinking water facility, and combined WASH facilities. Significant explanatory variables found in the bivariate analysis have been considered as explanatory variables in GLMMs. The odds ratios (OR), 95% confidence intervals (CI), and p-values obtained from mixed-effect logistic regression for analyzing individual and combined basic WASH facilities are presented in Table 5.

For basic handwashing facility, ICC = 0.117 indicates a moderate level of similarity within clusters. This means that about 11.7% of the total variation in handwashing facility availability can be attributed to differences between clusters, while the remaining 88.3% is due to individual differences within clusters. This suggests that there is some degree of clustering in the data, which should be accounted for in the analysis to obtain accurate estimates. Age of household head, partner's education level, media exposure, women empowerment, wealth index, mother's age, and mother's education of children under 5 have been found to have a significant association with basic handwashing facility. The households with household head aged between 40–49 years and 50–59 years had respectively 23.0% and 45.2% higher odds of having basic handwashing facility compared to the households with household head aged less than 30 years. In case of partner's education level, the study shows that those who had received higher level of education had 67.0% higher odds of having

**Table 5. Odds ratios (OR), 95% confidence intervals (CI) and p-values obtained from mixed effect logistic regression for analyzing individual and combined basic WASH facilities.**

| Covariates | Handwashing Facility | | | Sanitation Facility | | | Drinking Water Facility | | | Combined WASH Facilities | | |
|---|---|---|---|---|---|---|---|---|---|---|---|---|
| | OR | p-value | 95% CI | OR | p-value | 95% CI | OR | p-value | 95% CI | OR | p-value | 95% CI |
| **Intercept** | 0.324 | **<0.001** | (0.256, 0.409) | 0.261 | **<0.001** | (0.195, 0.349) | 18142.3 | **<0.001** | (4813.020, 68385.767) | 0.068 | **<0.001** | (0.053, 0.087) |
| **Division** | | | | | | | | | | | | |
| Barisal (Ref.) | – | – | – | – | – | – | – | – | – | – | – | – |
| Chittagong | – | – | – | 0.643 | **0.001** | (0.490, 0.842) | – | – | – | – | – | – |
| Dhaka | – | – | – | 0.339 | **<0.001** | (0.260, 0.443) | – | – | – | – | – | – |
| Khulna | – | – | – | 0.522 | **0.011** | (0.396, 0.688) | – | – | – | – | – | – |
| Mymensingh | – | – | – | 0.505 | **0.006** | (0.380, 0.671) | – | – | – | – | – | – |
| Rajshahi | – | – | – | 0.582 | **<0.001** | (0.442, 0.767) | – | – | – | – | – | – |
| Rangpur | – | – | – | 0.592 | **<0.001** | (0.449, 0.781) | – | – | – | – | – | – |
| Sylhet | – | – | – | 0.520 | **0.007** | (0.388, 0.696) | – | – | – | – | – | – |
| **Place of Residence** | | | | | | | | | | | | |
| Urban (Ref.) | – | – | – | – | – | – | – | – | – | – | – | – |
| Rural | 0.892 | 0.111 | (0.776, 1.025) | 1.578 | **<0.001** | (1.359, 1.831) | – | – | – | 1.040 | 0.588 | (0.903, 1.197) |
| **Sex of Household Head** | | | | | | | | | | | | |
| Male (Ref.) | – | – | – | – | – | – | – | – | – | – | – | – |
| Female | 1.050 | 0.450 | (0.925, 1.193) | 1.220 | **0.003** | (1.070, 1.391) | – | – | – | 1.143 | **0.042** | (1.005, 1.301) |
| **Age of Household Head** | | | | | | | | | | | | |
| <30 (Ref.) | – | – | – | – | – | – | – | – | – | – | – | – |
| 30-39 | 1.108 | 0.133 | (0.972, 1.264) | 1.448 | **0.003** | (1.265, 1.657) | – | – | – | 1.405 | **0.013** | (1.213, 1.627) |
| 40-49 | 1.230 | **0.008** | (1.056, 1.433) | 1.919 | **<0.001** | (1.644, 2.241) | – | – | – | 1.595 | **<0.001** | (1.356, 1.877) |
| 50-59 | 1.452 | **0.020** | (1.232, 1.712) | 2.555 | **<0.001** | (2.159, 3.024) | – | – | – | 2.048 | **<0.001** | (1.724, 2.434) |
| ≥60 | 1.169 | 0.054 | (0.997, 1.370) | 2.413 | **<0.001** | (2.051, 2.840) | – | – | – | 1.747 | **<0.001** | (1.479, 2.064) |
| **Household Size** | | | | | | | | | | | | |
| ≤5 (Ref.) | – | – | – | – | – | – | – | – | – | – | – | – |
| >5 | 1.080 | 0.115 | (0.981, 1.189) | 1.480 | **<0.001** | (1.339, 1.636) | – | – | – | 1.262 | **0.008** | (1.145, 1.392) |
| **Partner's Education Level** | | | | | | | | | | | | |
| No Education (Ref.) | – | – | – | – | – | – | – | – | – | – | – | – |

*(Continued)*

| Covariates | Handwashing Facility | | | Sanitation Facility | | | Drinking Water Facility | | | Combined WASH Facilities | | |
|---|---|---|---|---|---|---|---|---|---|---|---|---|
| | OR | p-value | 95% CI | OR | p-value | 95% CI | OR | p-value | 95% CI | OR | p-value | 95% CI |
| Primary | 0.969 | 0.573 | (0.870, 1.080) | 1.027 | 0.630 | (0.921, 1.147) | 0.916 | 0.749 | (0.533, 1.573) | 1.031 | 0.607 | (0.917, 1.160) |
| Secondary | 1.077 | 0.212 | (0.957, 1.211) | 1.116 | 0.074 | (0.990, 1.258) | 0.750 | 0.368 | (0.400, 1.404) | 1.153 | **0.026** | (1.019, 1.304) |
| Higher | 1.670 | **<0.001** | (1.406, 1.985) | 2.036 | **<0.001** | (1.700, 2.438) | 1.939 | 0.169 | (0.755, 4.977) | 2.034 | **<0.001** | (1.718, 2.407) |
| **Working Status** | | | | | | | | | | | | |
| No (Ref.) | – | – | – | – | – | – | – | – | – | – | – | – |
| Yes | 0.967 | 0.311 | (0.878, 1.043) | 0.873 | **0.002** | (0.799, 0.953) | 0.768 | 0.237 | (0.496, 1.189) | 0.873 | **0.003** | (0.799, 0.953) |
| **Media Exposure** | | | | | | | | | | | | |
| Non-exposed (Ref.) | – | – | – | – | – | – | – | – | – | – | – | – |
| Exposed | 1.147 | **0.001** | (1.054, 1.248) | 1.066 | 0.150 | (0.976, 1.164) | 1.226 | 0.402 | (0.760, 1.978) | 1.176 | **<0.001** | (1.077, 1.284) |
| **Migration** | | | | | | | | | | | | |
| Non-migrant (Ref.) | – | – | – | – | – | – | – | – | – | – | – | – |
| Migrant | 0.901 | 0.151 | (0.783, 1.038) | – | – | – | – | – | – | 0.998 | 0.974 | (0.863, 1.154) |
| **Woman Empowerment** | | | | | | | | | | | | |
| No (Ref.) | – | – | – | – | – | – | – | – | – | – | – | – |
| Yes | 1.165 | **0.014** | (1.032, 1.316) | – | – | – | 1.156 | 0.589 | (0.682, 1.959) | 1.165 | 0.335 | (0.936, 1.212) |
| **Wealth Index** | | | | | | | | | | | | |
| Poor (Ref.) | – | – | – | – | – | – | – | – | – | – | – | – |
| Middle | 1.952 | **<0.001** | (1.763, 2.162) | 2.203 | **<0.001** | (1.982, 2.449) | 2.210 | **0.006** | (1.252, 3.902) | 2.522 | **<0.001** | (2.255, 2.820) |
| Rich | 5.888 | **<0.001** | (5.276, 6.572) | 7.308 | **<0.001** | (6.472, 8.253) | 2.111 | **0.011** | (1.191, 3.741) | 8.559 | **<0.001** | (7.639, 9.590) |
| **Children Aged 5 and Under in Household** | | | | | | | | | | | | |
| No (Ref.) | – | – | – | – | – | – | – | – | – | – | – | – |
| Yes | 1.061 | 0.184 | (0.973, 1.156) | 0.972 | 0.533 | (0.889, 1.064) | – | – | – | 1.034 | 0.471 | (0.944, 1.131) |
| **Mother's Age of Under 5 Children** | | | | | | | | | | | | |
| ≤30 (Ref.) | – | – | – | – | – | – | – | – | – | – | – | – |
| >30 | 1.182 | **0.002** | (1.061, 1.316) | 1.565 | **<0.001** | (1.405, 1.743) | – | – | – | 1.502 | **<0.001** | (1.346, 1.677) |
| **Mother's Education of Under 5 Children** | | | | | | | | | | | | |
| No Education (Ref.) | – | – | – | – | – | – | – | – | – | – | – | – |
| Primary | 1.051 | 0.427 | (0.929, 1.189) | 1.119 | 0.082 | (0.987, 1.268) | 1.202 | 0.543 | (0.665, 2.173) | 1.010 | 0.881 | (0.882, 1.156) |
| Secondary | 1.288 | **<0.001** | (1.127, 1.471) | 1.338 | **0.019** | (1.166, 1.534) | 1.791 | 0.074 | (0.946, 3.394) | 1.332 | 0.054 | (1.155, 1.537) |

*(Continued)*

**Table 5.** (Continued)

| Covariates | Handwashing Facility | | | Sanitation Facility | | | Drinking Water Facility | | | Combined WASH Facilities | | |
|---|---|---|---|---|---|---|---|---|---|---|---|---|
| | OR | p-value | 95% CI | OR | p-value | 95% CI | OR | p-value | 95% CI | OR | p-value | 95% CI |
| Higher | 1.990 | **<0.001** | (1.681, 2.355) | 1.809 | **<0.001** | (1.523, 2.150) | 1.721 | 0.193 | (0.760, 3.897) | 1.998 | **<0.001** | (1.681, 2.374) |
| **Variance Component** | 0.4357 | | | 0.4906 | | | 52.17 | | | 0.4512 | | |
| **ICC** | 0.117 | | | 0.130 | | | 0. 941 | | | 0.121 | | |

Ref. = Reference Category

basic handwashing facility compared to those with no education. The study has also found that those who were exposed to media had 14.7% higher odds of having basic handwashing facility compared to those who were not exposed to media. Women who were empowered had 16.5% higher odds of having basic handwashing facility compared to the women who were not empowered. Those who belonged to the middle class had 95.2% higher odds of having basic handwashing facility compared to the poor, while the rich had 488.8% higher odds of having basic handwashing facility compared to the poor. The mothers of children under 5 children aged more than 30 years had 18.2% higher odds of having basic hand-washing facility compared to the mothers who were aged 30 years or less. The mothers who had received secondary and higher level of education had, respectively, 28.8% and 99.0% higher odds of having basic handwashing facility compared to the mothers with no education.

For basic sanitation facility, ICC = 0.130, which indicates about 13.0% of the total variation in sanitation facility avail-ability can be attributed to differences between clusters, while the remaining 87.0% is due to individual differences within clusters. Table 5 demonstrates that division, place of residence, sex, and age of household head, household size, part-ner's education level, working status, wealth index, mother's age, and mother's education of under 5 children had a significant association with basic sanitation facility. Chittagong, Dhaka, Khulna, Mymensingh, Rajshahi, Rangpur, and Sylhet had respectively 35.7%, 66.1%, 47.8%, 49.5%, 41.8%, 40.8%, and 48.0% lower odds of having basic sanitation facility compared to Barisal. Those living in rural areas had 57.8% higher odds of having basic sanitation facility compared to those living in urban areas. The households with a female household head had 22.0% higher odds of having basic sanitation facility compared to those with a male household head. The households with household heads aged between 30–39, 40–49 years, 50–59 years, 60 years and above had respectively 44.8%, 91.9%, 155.5%, and 141.3% higher odds of having basic sanitation facility compared to the households with household heads aged less than 30 years. Table 5 clarifies that the households with more than five members had 48.0% higher odds of having basic sanitation facility com-pared to the households with five or fewer members. In case of partner's education level, the study shows that those who had received higher level of education had 103.6% higher odds of having basic sanitation facility compared to those with no education. The working respondents had 12.7% lower odds of having basic sanitation facility compared to the non-working respondents. Moreover, those who belonged to the middle class had 120.3% higher odds of having basic sanita-tion facility compared to the poor, while the rich had 630.8% higher odds of having basic sanitation facility compared to the poor. The mothers of under 5 children aged more than 30 years had 56.5% higher odds of having basic sanitation facility compared to the mothers who were 30 years old or less. The mothers who had received secondary and higher level of education had, respectively, 33.8% and 80.9% higher odds of having basic sanitation facility compared to the mothers with no education.

Table 5 reveals that ICC = 0.941 for basic drinking water facility, which suggests a very high level of similarity within clusters. This shows that while individual differences within clusters account for only 5.9% of the variance in the availability of basic drinking water facilities, cluster differences account for 94.1% of the total variance. It has been found that only the

wealth index had a significant association with having basic drinking water facility. Table 5 shows that those who belonged to the middle class had 121.0% higher odds of having basic drinking water facility compared to the poor, and the rich had 111.1% higher odds of having basic drinking water facility compared to the poor.

Table 5 suggests ICC = 0.121 for combined WASH facilities, which indicates a moderate level of similarity within clusters. This means that about 12.1% of the total variation in combined WASH facilities availability can be attributed to differences between clusters, while the remaining 87.9% is due to individual differences within clusters. It can be seen from Table 5 that the sex and age of the household head, household size, partner's education level, working status, media exposure, wealth index, mother's age, and mother's education of children under 5 had a significant effect on combined WASH facilities. The households with a female household head had 14.3% higher odds of having combined WASH facilities compared to those with a male household head. The households with household head aged between 30–39, 40–49 years, 50–59 years, 60 years and above had, respectively, 40.5%, 59.5%, 104.8%, and 74.7% higher odds of having combined WASH facilities compared to the households with household head aged less than 30 years. The households with more than five members had 26.2% higher odds of having combined WASH facilities compared to the households with 5 or fewer members. The study shows that those whose partners had received secondary and higher level of education had, respectively, 15.3% and 103.4% higher odds of having combined WASH facilities compared to those whose partners were illiterate. The working respondents had 12.7% lower odds of having combined WASH facilities compared to the non-working respondents. The study has also found that those who were exposed to media had 17.6% higher odds of having combined WASH facilities compared to those who were not exposed to media. Those who belonged to the middle class had 152.2% higher odds of having combined WASH facilities compared to the poor, while the rich had 755.9% higher odds of having combined WASH facilities compared to the poor. The mothers of under 5 children aged more than 30 years had 50.2% higher odds of having combined WASH facilities compared to the mothers who were aged 30 years or less. The mothers who had received higher level of education had 99.9% higher odds of having combined WASH facilities compared to the mothers with no education.

## Mixed-effect vs. Fixed-effect logistic regression model

For analyzing individual and combined WASH facilities, the performances of the mixed effect logistic regression model (model 1) and the fixed effect logistic regression model (model 2) have been compared based on AIC values and by performing likelihood ratio test (LRT).

The AIC values as well as the log-likelihood values for the mixed-effect logistic regression model (model 1) and the fixed-effect logistic regression model (model 2), along with $\chi^2$-value and p-value, are presented in Table 6.

From Table 6, it has been found that the mixed-effect logistic regression model (model 1) has lower AIC value than the usual logistic regression model with fixed effects (model 2). This is an indication that the mixed-effect logistic regression

**Table 6. Performance comparison of mixed-effect logistic regression model (Model 1) and fixed-effect logistic regression model (Model 2) for individual and combined WASH facilities based on AIC values and LRT.**

| Facility | Model | AIC | Log-likelihood | $\chi^2$ | p-value |
|---|---|---|---|---|---|
| **Basic Handwashing Facility** | Model 1 | **17373.3** | −8663.7 | 473.15 | **<0.001** |
| | Model 2 | 17844.0 | −8900.0 | | |
| **Basic Sanitation Facility** | Model 1 | **16636.8** | −8290.4 | 534.33 | **<0.001** |
| | Model 2 | 17169.0 | −8557.5 | | |
| **Basic Drinking Water Facility** | Model 1 | **1166.4** | −570.2 | 861.97 | **<0.001** |
| | Model 2 | 2083.8 | −1029.9 | | |
| **Combined WASH Facilities** | Model 1 | **16483.4** | −8218.7 | 433.65 | **<0.001** |
| | Model 2 | 16915.0 | −8435.5 | | |

model seems to be a better fit for analyzing individual and combined WASH facilities since the model with a lower AIC value suggests a better fit for the data. The p-value has been obtained for the null hypothesis that the fixed-effect logistic regression model performs equally well as the mixed-effect logistic regression model. Since the p-value is less than 0.001, the null hypothesis can be rejected, which reveals that the mixed-effect model outperforms the fixed-effect model significantly. This also indicates that the mixed-effect logistic regression model offers a substantially better fit to the data than the fixed-effect logistic regression model for analyzing individual and combined WASH facilities. Hence, the use of the mixed-effect logistic regression model is appropriate.

## Discussion

Traditional logistic regression models are not suitable for analyzing clustered data, as responses within the same cluster are not independent, despite clusters being independent of each other. To account for this dependency, a GLMM is necessary for obtaining consistent and efficient estimators for the regression parameters of interest. This paper reveals several key determinants and barriers to WASH facilities in Bangladesh using clustered data from BDHS 2022, employing a GLMM for binary response, which is also known as mixed-effect logistic regression model. Based on AIC values and LRT, this study has found that the mixed effect logistic regression model outperforms the fixed effect logistic regression model. This justifies the adoption of the mixed-effect logistic regression model. Also, the existence of correlation within the same cluster has been justified by calculating ICC while analyzing individual and combined basic WASH facilities.

Among all the divisions, Barisal was found to have a higher likelihood of possessing a basic sanitation facility, a finding that remains consistent with a previous study [18]. This reinforces the critical role that regional factors, possibly related to local governance, cultural norms, or targeted public health initiatives, play in shaping sanitation infrastructure. The study also confirms earlier results by demonstrating that the wealth index is positively associated with both individual and combined basic WASH facilities [15–17,19,22,25]. This economic influence suggests that households with greater financial resources are better positioned to secure the necessary WASH facilities, underscoring the importance of promoting economic development as part of public health strategies.

Media exposure emerged as another significant determinant, showing a positive association with basic handwashing and combined WASH facilities. This implies that effective public health campaigns and media outreach can raise awareness and drive the adoption of healthier practices. In contrast, working status was negatively associated with both basic sanitation and combined WASH facilities, which may reflect the impact of occupational constraints or time limitations on maintaining critical household services. Additionally, the positive influence of the household head's age on basic handwashing, sanitation, and combined WASH facilities suggests that experience and household stability are important factors in prioritizing and maintaining these amenities. The analysis further indicates that female-headed households and those with more than five members are more likely to have basic sanitation and combined WASH facilities, potentially due to differing household priorities or the ability to pool resources effectively within larger family structures. These findings align with previous studies [15–18,24,25].

Interestingly, rural residents were also more likely to have basic sanitation facilities hinting at the possibility that community-based local initiatives in rural areas might be more successful or that rural settings allow for better adaptation of certain sanitation practices. While this finding is consistent with a previous study [15], it also stands in contrast to several earlier studies [16,17,25]. This contradiction could be attributed to differences in research methods. Moreover, shifts in policy and local engagement over time might influence these findings, especially since recent improvements in rural sanitation may not have been reflected in earlier studies.

Partner's education level, along with the age and education of mothers of children under 5, played a pivotal role across all aspects of WASH facilities. Households with partners or mothers possessing lower or no education were less likely to have these facilities, underscoring how the educational environment within a household can significantly affect health

behaviors and resource prioritization [27]. Moreover, the observed positive correlation between basic handwashing facilities and women's empowerment further reinforces the idea that empowering women, concerning decision-making and resource management, leads to improvements in critical hygiene practices [28].

Together, these findings highlight a multifaceted landscape where economic, educational, regional, and gender-related factors converge to influence WASH outcomes in Bangladesh. The evidence suggests that policy interventions must be both targeted and comprehensive, promoting economic uplift, investing in educational and media initiatives, and tailoring strategies to the unique needs of various regions and household structures. By addressing these determinants holistically, policymakers can better combat waterborne diseases, improve overall public health, and work toward sustainable and equitable access to essential WASH services.

## Conclusion

This study reveals that various factors may influence the effectiveness of WASH practices in Bangladesh, which are vital for mitigating waterborne diseases and ensuring equitable access to services. The analysis indicates that enhanced handwashing facilities tend to be associated with higher levels of partner and maternal education, increased media exposure, and improved women's empowerment—findings that underscore the potential benefits of targeted public health and educational initiatives. Furthermore, the significant regional, household, and economic disparities observed in sanitation suggest that localized policy measures, such as municipal reforms and targeted financial support, might lead to improved sanitation infrastructure and practices. The correlation between lower wealth levels and reduced access to clean drinking water implies that policies involving direct financial aid, infrastructural investments, or subsidy programs could be particularly beneficial for economically disadvantaged areas. Additionally, the observed effectiveness of combined WASH facilities, especially in households headed by men, as well as in smaller and working-member households, highlights the potential importance of integrated policy strategies that further bolster education and media outreach. Together, these insights provide a practical framework for progressing toward sustainable improvements in WASH outcomes, thereby advancing public health and promoting fair service delivery.

## Limitations and Further Scopes

This study utilizes cross-sectional data, which may introduce selection and information biases, potentially affecting the accuracy of observed associations. Since the data represents a single point in time, it limits the ability to analyze longitudinal trends and establish causal relationships, restricting insights into the evolution of WASH facilities and their long-term impacts. Additionally, while efforts were made to ensure representativeness, generalizability may be limited due to regional variations in governance, infrastructure, and cultural practices.

To enhance future research, longitudinal studies should be undertaken to track temporal changes in WASH accessibility and strengthen causal inference. Furthermore, incorporating qualitative assessments could provide deeper insights into social accountability mechanisms such as incentives, sanctions, and responsiveness, helping to understand community engagement with WASH services. Exploring additional socioeconomic and environmental determinants influencing access would refine the analysis, and assessing the effectiveness of behavior change interventions, including hygiene awareness programs, financial subsidies, and infrastructure development, could lead to more practical policy recommendations for sustainable WASH improvements.

## Acknowledgments

The authors are grateful to the Demographic and Health Survey (DHS) for executing a nationwide survey and making its data publicly available.

## Author contributions

**Conceptualization:** Mahmila Sanjana Mim, Lutfor Rahaman.

**Data curation:** Mahmila Sanjana Mim.

**Formal analysis:** Mahmila Sanjana Mim.

**Methodology:** Mahmila Sanjana Mim, Anamul Sajib.

**Software:** Mahmila Sanjana Mim.

**Supervision:** Lutfor Rahaman, Anamul Sajib.

**Writing – original draft:** Mahmila Sanjana Mim.

**Writing – review & editing:** Lutfor Rahaman, Anamul Sajib.

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
