## [Decision Letter · Decision Letter 0]

Dear Dr. Mim,

Thank you for submitting your manuscript to PLOS ONE. After careful consideration, we feel that it has merit but does not fully meet PLOS ONE’s publication criteria as it currently stands. Therefore, we invite you to submit a revised version of the manuscript that addresses the points raised during the review process.

We look forward to receiving your revised manuscript.

Kind regards,

Md. Moyazzem Hossain, PhD

Academic Editor

PLOS ONE

Journal Requirements:

Reviewers' comments:

Reviewer's Responses to Questions

**Comments to the Author**

1. Is the manuscript technically sound, and do the data support the conclusions?

Reviewer #1: Yes

Reviewer #2: Yes

Reviewer #3: No

2. Has the statistical analysis been performed appropriately and rigorously?

Reviewer #1: Yes

Reviewer #2: Yes

Reviewer #3: No

3. Have the authors made all data underlying the findings in their manuscript fully available?

Reviewer #1: Yes

Reviewer #2: Yes

Reviewer #3: No

4. Is the manuscript presented in an intelligible fashion and written in standard English?

Reviewer #1: Yes

Reviewer #2: Yes

Reviewer #3: No

Reviewer #1: Dear Authors,

The article is very interesting in exploring determinants of WSH facilities ownership in Bangladesh using GLMM analysis. I understand the that article have important contribution in regards of the analytical methodology, however, the reader also expect thorough discussion in regards to the findings on the significant determinants.

One of interesting findings is that respondents living in rural areas has higher chance to have basic sanitation facilities compare to urban areas. The author shows that similar finding from other studies, but it is worth to discuss to other studies that have contradicting findings. This could be provide detailed understanding on urban-rural discrepancy and reshape the recommendation for improving WASH service provision.

Thank you

Reviewer #2: echnical Soundness and Data Support: As stated in the introduction, the manuscript is complete from a technical point of view because it applied a Generalized Linear Mixed Model (GLMM) to account for intra cluster correlation in the BDHS 2022 data. The analysis of data demonstrated regionally WASH facilities in Bangladesh and the conclusions drawn were logic.

Statistical Analysis: As mentioned previously, the statistical analysis is sophisticated, and for this task appropriate. GLMM was preferred to fixed-effect logistic regression as revealed by the AIC and LRT, ensuring correct inferences will be made with the clustered data.

Data Availability: The data are available on the DHS website, therefore meeting fully accessible criteria for no restrictions of use.

Language and Presentation: The text is very coherent and presented in standard English. Yet, small mistakes in typing and grammar should be amended in the revision.

Additional Comments:

The study is important in understanding WASH facilities in Bangladesh as the country strives to achieve the 6th SDG, and thus offers relevant contribution.

Try to include more specific implications concerning the policies in the findings to increase the practical effect.

Reviewer #3: The manuscript “Generalized Linear Mixed Model Approach for Analyzing Water, Sanitation, and Hygiene Facilities in Bangladesh: Insights from BDHS 2022 Data” presents a significant global health problem. While the topic is relevant, the manuscript requires significant revisions before it can be considered for publication.

Major concerns:

1. While the machine learning model is a powerful algorithm, the manuscript lacks a clear justification for why the GLM method was chosen over other potential models. A more thorough comparison of different models and their suitability for this specific problem is needed.

2. The description of the data preprocessing, model selection, and evaluation methods needs to be significantly improved. There are vague statements and missing details.

3. This manuscript will benefit from spatial analysis with district-level data.

4. The manuscript does not adequately address the limitations of the study, such as potential biases in the data, the challenges of causal inference, and the generalizability of the findings.

5. The interpretation of the results sometimes appears overstated. The manuscript should be more cautious in drawing conclusions and acknowledge the uncertainties associated with the model predictions.

6. The figures and tables need to be improved in terms of clarity and presentation.

7. There are some inconsistencies in the referencing style.

8. The abstract is a bit too long and could be more concise.

Specific Points:

Methods:

1. Provide more details about the data sources, including specific databases and data collection procedures.

2. Clearly explain the data preprocessing steps, including how missing values were handled, how outliers were identified and treated, and how variables were standardized or normalized.

Results:

1. Present the results of the model evaluation clearly and concisely. Include tables or figures showing the performance metrics for each model.

2. Provide more details about the key factors identified by the SHAP analysis (proposed). Discuss the potential mechanisms through which these factors influence Water, Sanitation, and Hygiene Facilities in Bangladesh.

3. Ensure that all tables and figures are properly labeled and referenced in the text.

Discussion:

1. Discuss the limitations of the study in more detail. Acknowledge potential biases in the data, the challenges of causal inference, and the generalizability of the findings.

2. Compare the findings of this study with those of previous research. Discuss the similarities and differences and explain why the results might differ.

3. Provide more specific recommendations for public health interventions based on the findings of the study.

Abstract:

1. Make the abstract more concise and focused. Highlight the key findings and their implications.

**Do you want your identity to be public for this peer review?** For information about this choice, including consent withdrawal, please see our Privacy Policy

Reviewer #1: **Yes: ** Ni Made Utami Dwipayanti

Reviewer #2: **Yes: ** Hala Awad Ahmed

Reviewer #3: **Yes: ** Md. Siddikur Rahman

---

## [Author Response · Author response to Decision Letter 1]

15 May 2025

PONE-D-25-02295

Generalized Linear Mixed Model Approach for Analyzing Water, Sanitation, and Hygiene Facilities in Bangladesh: Insights from BDHS 2022 Data

I appreciate the feedback provided by the reviewers for our manuscript titled “Generalized Linear Mixed Model Approach for Analyzing Water, Sanitation, and Hygiene Facilities in Bangladesh: Insights from BDHS 2022 Data”.

Answer to the Reviewers' comments:

1. Is the manuscript technically sound, and do the data support the conclusions?

Reviewer #1: Yes

Reviewer #2: Yes

Reviewer #3: No

Answer: Thank you for acknowledging the technical soundness of our research. The primary contribution of our work lies in the investigation of key determinants of WASH facilities in Bangladesh by following GLMM approach. We have rigorously conducted the experiments, and the conclusions are drawn appropriately based on the data. However, the third reviewer disagreed. Considering the third reviewer’s comment, we have modified our “Conclusion” section so that it looks more appropriate.________________________________________

2. Has the statistical analysis been performed appropriately and rigorously?

Reviewer #1: Yes

Reviewer #2: Yes

Reviewer #3: No

Answers: We sincerely appreciate the first and second reviewer for recognizing the rigor and appropriateness of our statistical analysis. We also appreciate the disagreement of the third reviewer and his recommendation regarding the use of machine learning approach. However, our aim was to conduct the analysis under parametric framework. Performing machine learning analysis could be a future work.

3. Have the authors made all data underlying the findings in their manuscript fully available?

Reviewer #1: Yes

Reviewer #2: Yes

Reviewer #3: No

Answer: Thank you so much. All data underlying the findings described in the manuscript are fully available on DHS website without any restriction.

4. Is the manuscript presented in an intelligible fashion and written in standard English?

Reviewer #1: Yes

Reviewer #2: Yes

Reviewer #3: No

Answer: Thank you very much. We have carefully reviewed the manuscript and corrected grammatical errors.

5. Review Comments to the Author

Reviewer #1: Dear Authors,

The article is very interesting in exploring determinants of WSH facilities ownership in Bangladesh using GLMM analysis. I understand that the article have important contribution in regards of the analytical methodology, however, the reader also expect thorough discussion in regards to the findings on the significant determinants. One of interesting findings is that respondents living in rural areas has higher chance to have basic sanitation facilities compare to urban areas. The author shows that similar finding from other studies, but it is worth to discuss to other studies that have contradicting findings. This could be provide detailed understanding on urban-rural discrepancy and reshape the recommendation for improving WASH service provision.

Thank you.

Answer: Thank you very much for your valuable comment. We have made thorough discussion in regards to the findings on the significant determinants according to your comment. Also, a discussion on the contradicting findings from other studies has been made in the “Discussion” section on page 30.

Reviewer #2: Technical Soundness and Data Support: As stated in the introduction, the manuscript is complete from a technical point of view because it applied a Generalized Linear Mixed Model (GLMM) to account for intra cluster correlation in the BDHS 2022 data. The analysis of data demonstrated regionally WASH facilities in Bangladesh and the conclusions drawn were logic.

Statistical Analysis: As mentioned previously, the statistical analysis is sophisticated, and for this task appropriate. GLMM was preferred to fixed-effect logistic regression as revealed by the AIC and LRT, ensuring correct inferences will be made with the clustered data.

Data Availability: The data are available on the DHS website, therefore meeting fully accessible criteria for no restrictions of use.

Language and Presentation: The text is very coherent and presented in standard English. Yet, small mistakes in typing and grammar should be amended in the revision.

Answer: Thank you so much for such encouraging words. We have carefully revised the manuscript and corrected the typing mistakes and grammatical errors.

Additional Comments:

The study is important in understanding WASH facilities in Bangladesh as the country strives to achieve the 6th SDG, and thus offers relevant contribution.

Try to include more specific implications concerning the policies in the findings to increase the practical effect.

Answer: Thank you very much. We have included more specific implications according to your recommendation in the “Conclusion” section on page 31.

Reviewer #3: The manuscript “Generalized Linear Mixed Model Approach for Analyzing Water, Sanitation, and Hygiene Facilities in Bangladesh: Insights from BDHS 2022 Data” presents a significant global health problem. While the topic is relevant, the manuscript requires significant revisions before it can be considered for publication.

Major concerns:

1. While the machine learning model is a powerful algorithm, the manuscript lacks a clear justification for why the GLM method was chosen over other potential models. A more thorough comparison of different models and their suitability for this specific problem is needed.

Answer: Thank you very much for your valuable comment. We really appreciate your recommendation. Our aim was to conduct the analysis under parametric (classical) framework. We wanted to take into account the cluster effect which exists in the dataset that we used. The reason behind the application of the mixed-effect logistic regression model (GLMM) has also been mentioned in the manuscript on page 4. Moreover, a comparison with fixed-effect logistic regression model (GLM) has been made and hence, the appropriateness of using the GLMM over the GLM has been justified. The comparison can be found in the manuscript on page 28.

2. The description of the data preprocessing, model selection, and evaluation methods needs to be significantly improved. There are vague statements and missing details.

Answer: Thank you so much. We have improved the description of the data processing method in “Statistical Analyses” section on page 9. Also, the model selection, and evaluation methods have been improved in the “Mixed-Effect vs. Fixed Effect Logistic Regression Model” section on the pages 27 and 28.

3. This manuscript will benefit from spatial analysis with district-level data.

Answer: Thank you very much for your thoughtful suggestion regarding the incorporation of spatial analysis using district-level data. We truly appreciate the insight, and I agree that this approach could provide valuable localized insights into geographic disparities in WASH facilities. However, given the current scope and focus of this study, we have chosen not to include spatial analysis at this time. We intend to explore this promising avenue in future research, as it would allow for a more detailed examination of district-level variations and further enhance policy recommendations. Thank you again for your valuable input.

4. The manuscript does not adequately address the limitations of the study, such as potential biases in the data, the challenges of causal inference, and the generalizability of the findings.

Answer: Thank you so much. We have revised the manuscript and addressed the limitations adequately in the “Limitations and Further Scopes” section on page 32.

5. The interpretation of the results sometimes appears overstated. The manuscript should be more cautious in drawing conclusions and acknowledge the uncertainties associated with the model predictions.

Answer: Thank you very much. We have modified the conclusion section based on your valuable suggestion.

6. The figures and tables need to be improved in terms of clarity and presentation.

Answer: Thank you very much. As per your recommendation, we have improved the tables.

7. There are some inconsistencies in the referencing style.

Answer: Thank you very much. Referencing styles have been standardized. APA referencing style has been adhered.

8. The abstract is a bit too long and could be more concise.

Answer: Thank you so much. We have concised the abstract.

Specific Points:

Methods:

1. Provide more details about the data sources, including specific databases and data collection procedures.

Answer: Thank you so much. More details about the data sources, including specific databases and data collection procedures have been provided in the “Data Sources” section on page 5 in the revised manuscript.

2. Clearly explain the data preprocessing steps, including how missing values were handled, how outliers were identified and treated, and how variables were standardized or normalized.

Answer: Thank you very much. According to your valuable comment, a clear explanation regarding the data preprocessing steps has been provided on page 9 in the “Statistical Analyses” section.

Results:

1. Present the results of the model evaluation clearly and concisely. Include tables or figures showing the performance metrics for each model.

Answer: Thank you so much. We have presented the results of the model evaluation clearly and concisely in the revised manuscript.

2. Provide more details about the key factors identified by the SHAP analysis (proposed). Discuss the potential mechanisms through which these factors influence Water, Sanitation, and Hygiene Facilities in Bangladesh.

Answer: Thank you very much. We have not considered SHAP analysis in our study. We may explore this in future research. But we have provided more details about the key factors identified by GLMM approach and discussed the potential mechanisms through which these factors influence WASH Facilities in Bangladesh.

3. Ensure that all tables and figures are properly labeled and referenced in the text.

Answer: Thank you so much. In the revised manuscript, we have ensured that all tables and figures are properly labeled and referenced in the text.

Discussion:

1. Discuss the limitations of the study in more detail. Acknowledge potential biases in the data, the challenges of causal inference, and the generalizability of the findings.

Answer: Thank you very much. A detailed discussion has been made regarding the limitations of the study according to your recommendation.

2. Compare the findings of this study with those of previous research. Discuss the similarities and differences and explain why the results might differ.

Answer: Thank you so much. According to your valuable comment, we have compared the findings of this study with those of previous research in the “Discussion” section on the pages 29 and 30 in the revised manuscript.

3. Provide more specific recommendations for public health interventions based on the findings of the study.

Answer: Thank you very much. In the revised manuscript, we have provided more specific recommendations for public health interventions based on the findings of the study.

Abstract:

1. Make the abstract more concise and focused. Highlight the key findings and their implications.

Answer: Thank you so much. We have made the abstract more concise and focused in the revised manuscript.

6. PLOS authors have the option to publish the peer review history of their article (what does this mean?). If published, this will include your full peer review and any attached files.

Do you want your identity to be public for this peer review? For information about this choice, including consent withdrawal, please see our Privacy Policy.

Reviewer #1: Yes: Ni Made Utami Dwipayanti

Reviewer #2: Yes: Hala Awad Ahmed

Reviewer #3: Yes: Md. Siddikur Rahman

---

## [Decision Letter · Decision Letter 1]

Generalized Linear Mixed Model Approach for Analyzing Water, Sanitation, and Hygiene Facilities in Bangladesh: Insights from BDHS 2022 Data

PONE-D-25-02295R1

Dear Dr. Mim,

We’re pleased to inform you that your manuscript has been judged scientifically suitable for publication and will be formally accepted for publication once it meets all outstanding technical requirements.

Kind regards,

Md. Moyazzem Hossain, PhD

Academic Editor

PLOS ONE

Additional Editor Comments (optional):

Reviewers' comments:

Reviewer's Responses to Questions

**Comments to the Author**

Reviewer #1: All comments have been addressed

Reviewer #2: All comments have been addressed

2. Is the manuscript technically sound, and do the data support the conclusions?

Reviewer #1: Yes

Reviewer #2: Yes

3. Has the statistical analysis been performed appropriately and rigorously?

Reviewer #1: Yes

Reviewer #2: Yes

4. Have the authors made all data underlying the findings in their manuscript fully available?

Reviewer #1: Yes

Reviewer #2: Yes

5. Is the manuscript presented in an intelligible fashion and written in standard English?

Reviewer #1: Yes

Reviewer #2: Yes

Reviewer #1: Thank you for responding my comments and revise accordingly. To my understanding, the bivariate analysis actually showed a different result, that urban dweller is more likely to have access to handwashing facility and basic sanitation access. But yes, different methods of analysis might lead to different results

Reviewer #2: The revised manuscript titled "Generalized Linear Mixed Model Approach for Analyzing Water, Sanitation, and Hygiene Facilities in Bangladesh: Insights from BDHS 2022 Data" demonstrates considerable improvement over the previous version. The authors have addressed the reviewers' concerns with care and thoroughness. The justification for choosing the Generalized Linear Mixed Model (GLMM) over other modeling approaches has been clearly explained and is appropriate given the clustered nature of the BDHS data. Statistical analysis is methodologically sound, appropriately applied, and well-supported by the data.

The revisions to the manuscript’s structure, language, and clarity have made the content more intelligible and accessible. Typographical and grammatical errors noted previously have been corrected. The data availability statement complies with PLOS ONE’s policy, with the dataset being accessible on the DHS website.

The authors have strengthened the discussion of their findings, acknowledged limitations, and expanded the policy implications to enhance practical relevance. While suggestions such as spatial analysis and SHAP were acknowledged as future work, the current scope remains valid and valuable for publication.

I recommend accepting this manuscript for publication.

**Do you want your identity to be public for this peer review?** For information about this choice, including consent withdrawal, please see our Privacy Policy

Reviewer #1: **Yes: ** Ni Made Utami Dwipayanti

Reviewer #2: **Yes: ** Hala Awad Ahmed

---

## [Editor Report · Acceptance letter]

PONE-D-25-02295R1

PLOS ONE

Dear Dr. Sajib,

I'm pleased to inform you that your manuscript has been deemed suitable for publication in PLOS ONE. Congratulations! Your manuscript is now being handed over to our production team.

Kind regards,

on behalf of

Professor Md. Moyazzem Hossain

Academic Editor

PLOS ONE